# Effect of UV Radiation on Structural Damage and Tribological Properties of Mo/MoS₂-Pb-PbS Composite Films

Cuihong Han [1,2], Guolu Li [1,*], Guozheng Ma [3,*], Jiadong Shi [1], Zhen Li [4], Qingsong Yong [5] and Haidou Wang [3]

1   School of Materials Science and Engineering, Hebei University of Technology, Tianjin 300401, China; hanmutou@163.com (C.H.); jiadong1207@126.com (J.S.)
2   School of Mechanical Engineering, Tianjin University of Technology and Education, Tianjin 300222, China
3   National Key Laboratory for Remanufacturing, Army Academy of Armored Forces, Beijing 100072, China; wanghaidou@tsinghua.org.cn
4   State Key Laboratory of Mechanical Systems and Vibrations, Shanghai Jiao Tong University, Shanghai 200240, China; Lizhen2019@sjtu.edu.cn
5   China Aerodynamics Research and Development Center, Mianyang 621000, China; qingsong_yong@163.com
*   Correspondence: liguolu0305@163.com (G.L.); magz0929@163.com (G.M.); Tel.: +86-22-60202012 (G.L.); +86-10-66718475 (G.M.); Fax: +86-10-66717144 (G.M.)

**Abstract:** To investigate ultraviolet (UV) radiation effects on tribological properties of Mo/MoS₂-Pb-PbS film, ultraviolet (UV) radiation exposure tests were carried out for 20 h, 40 h, 60 h and 80 h by space UV radiation simulation device developed by our team, which can reach 3 UV radiation intensity. The exposure time in test was equivalent to the radiation of 100 h, 200 h, 300 h and 400 h in the space. Then, the vacuum friction test of Mo/MoS₂-Pb-PbS thin film was performed under the 6 N load and 100 r/min, and friction test time of each sample was 20 min. By SEM, TEM, XPS the composition and morphology of Mo/MoS₂-Pb-PbS film surface after UV radiation were analyzed. UV radiation could change the microstructure significantly and relative content of S element and MoS₂ on the surface of the films decreased, and light mass loss of the films occurred. The tribological properties will also recover with the increase of sliding time, although the friction coefficient fluctuation of the film increased at the starting stage of the friction test. The damage of Mo/MoS₂-Pb-PbS under UV irradiation was mainly caused by the volatilization of the enriched S element in the surface layer due to the high temperature heating of UV irradiation.

**Keywords:** UV radiation; Mo/MoS₂-Pb-PbS composite film; tribological properties; structural damage

## 1. Introduction

Molybdenum disulfide is widely used in the lubrication field of space equipment due to its good properties such as high and low temperature resistance, radiation resistance, and vacuum cold welding [1–3]. Space equipment will be subjected to intense solar radiation during on-orbit operation. In the solar radiation spectrum, the energy provided by ultraviolet (UV) radiation with wavelength of 100–400 nm is about 12–3 eV, while the first ionization energies of S element and Mo element in MoS₂ coatings are 10.360 electronic volts and 7.099 eV, respectively [4]. Under the condition of space UV radiation, the element composition and microstructure of high-energy UV photons change correspondingly, which affects the lubrication performance and lubrication life. It was found that the effect of vacuum ultraviolet radiation on the life of MoS₂ solid lubrication coating was mainly related to the composition of the coating [5]. For MoS₂/Si coating, vacuum ultraviolet radiation reduces its life by about 38.9%. Under the same radiation conditions, the lifetime of MoS₂/epoxy coating decreased only about 8.8% [6]. Yan et al. studied the changes of electrode materials, especially the interface structure of thin films which formed electronic components under UV radiation. It was found UV radiation accelerated the diffusion rate of Cu atoms to the surface and crystal boundary of Au layer. With the extension of radiation time, the

promoting effect gradually tended to be flat [7]. Hexamethyldisiloxane modified $SiO_2$ with high stability and easy dispersion in organic solvent was added to $MoS_2$/phenolic epoxy resin bonded solid lubricant coating by Liyan [8]. The ground simulation experiment conditions of ultraviolet radiation were as follows: vacuum degree was less than or equal to $10^{-4}$ Pa, wavelength range was 115–400 nm, average radiation intensity was 700 W/m$^2$ (about six ultraviolet solar constants), radiation time was 10 h. When the radiation dose is about 60 equivalent sunlight (esh) ultraviolet radiation is beneficial to improving the wear resistance of the coating to a certain extent. The addition of $SiO_2$ significantly improves the tribological properties and ultraviolet radiation resistance of the composite coating. For these reasons as well as proven operation in harsh environments, the unique properties of $MoS_2$ motivated the study of its suitability as a material in space electronics applications [9,10], Julian J. McMorrow, etc., quantified the response of $MoS_2$ field effect transistors (FETs) to vacuum ultraviolet (VUV) total ionizing dose radiation. Single-layer and multilayer $MoS_2$ FETs were compared to identify differences that arise from thickness and band structure variations. However, Raman spectroscopy showed no variation in the $MoS_2$ signatures as a result of VUV exposure, eliminating significant crystalline damage or oxidation as possible radiation degradation mechanisms [11]. Thereby, it is necessary to study the friction performance of molybdenum disulfide composite film under UV radiation, in order to enlarge application of $MoS_2$ composite film in space equipment furtherly. In a word, UV irradiation energy may break the binding bond of the compound and change the material properties especially the self-lubricating material used by space equipment.

In this paper, the soft metal lead (Pb) was doped into the metal molybdenum (Mo) film by the composite surface engineering technology of 'PVD coating + low temperature ion sulfurization', and the Mo/$MoS_2$-Pb-PbS multiple composite solid lubricant film was prepared by low temperature ion sulfurization process. Mo/$MoS_2$-Pb-PbS composite films were exposed to UV radiation environment, and then the high vacuum friction test was carried out to study the microstructure and tribological properties of Mo-based multicomponent composite solid lubricating film before and after UV radiation. The damage mechanism of the film after UV radiation was briefly analyzed.

## 2. Experimental Details

### 2.1. Materials and Equipment

The substrate of Mo/$MoS_2$-Pb-PbS composite film was 9Cr18 (AISI440C) bearing steel, and the sample was a disk of $\Phi$34.0 mm with thickness of 3 mm. The preparation of Mo/$MoS_2$-Pb-PbS composite films were completed by magnetron sputtering and low-temperature plasma sulfurization. The specific preparation process, parameter and film structure can be referred to reference [9]. The film thickness was about 1900 nm, and it showed good tribological properties in vacuum and atomic oxygen environment.

The UV radiation test of Mo/$MoS_2$-Pb-PbS composite film was test by space UV radiation simulation device developed by our team, which was optimized and integrated with the MSTS-1 vacuum friction and wear tester (MSTS-1, Beijing, China). The schematic diagram of the simulation device was shown in Figure 1. Two light sources were used to simulate near ultraviolet radiation and medium ultraviolet radiation by space UV radiation simulation device. The short wavelength vacuum ultraviolet cannot penetrate the quartz window and was easily absorbed by air, so the method of hanging deuterium lamp directly in the vacuum chamber and near the sample table was used for simulation. The near ultraviolet radiation device was simulated by special light source and optical path transformation mechanism. The light emitted by the two light sources was not a linear spectrum, and the middle superposition area can completely cover the ultraviolet spectrum. The technical indicators of space UV radiation simulation device were shown in Table 1.

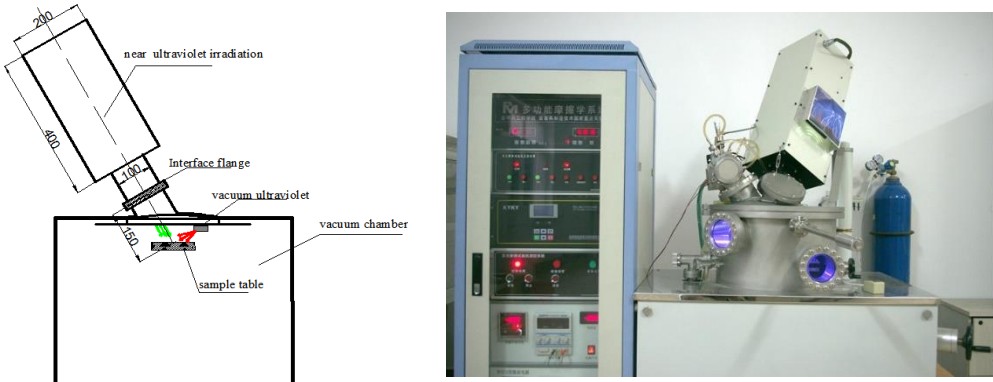

**Figure 1.** The schematic diagram of the UV radiation simulation device and MSTS-1 vacuum friction and wear tester integrated with UV radiation simulation device.

**Table 1.** Technical parameters of space UV radiation simulation device.

| Parameter | Value | Parameter | Value | Parameter | Value |
|---|---|---|---|---|---|
| UV radiation intensity collimation angle | >3 UV constants <5° | spectral range UV radiation uniformity | 120–400 nm >±10% | radiation area stability | >Φ50 mm >5%/h |

After the sample was exposed by UV radiation, the MSTS-1 multi-functional vacuum friction and wear tester was used to test the vacuum tribological performance. The structural schematic diagram of the MSTS-1 vacuum friction and wear tester is shown in References [12,13]. In the friction test, the grinding ball was 9Cr18 bearing steel with Φ9.525 mm and G10 grade, which hardness was HRC58 and surface roughness was Ra 0.025 μm. The following samples were Mo/MoS$_2$-Pb-PbS composite film disk samples (Φ34 mm × 3 mm) exposed in UV radiation; the friction track diameter of the upper sample steel ball and the lower disc sample is 20 mm. During the test, the vacuum chamber pressure of the friction testing machine is maintained at $8 \times 10^{-5}$ Pa.

### 2.2. Process

In this paper, the UV exposure test of the Mo/MoS$_2$-Pb-PbS composite film was carried out using the developed space UV radiation simulation device. There are eight samples exposed in UV environment, two samples as a group. The exposure time of each group was setting as 20 h, 40 h, 60 h and 80 h in order, which was equivalent to the radiation of the actual space exposure for 100 h, 200 h, 300 h and 400 h.

Then, the friction test of Mo/MoS$_2$-Pb-PbS film was measured by MSTS-1 multi-functional vacuum friction and wear tester under high vacuum ($8 \times 10^{-5}$ Pa), 6 N and 100 r/min. If the friction coefficient continued to exceed 0.6 during the test, the test was stopped, otherwise it was continuously operated for 20 min. During the test, the friction coefficient was lower than 0.6, and the friction test time was 20 min.

In this paper, the microstructure changes of Mo/MoS$_2$-Pb-PbS composite films before and after UV radiation were compared with the samples after 40 h UV radiation.

### 3. Results and Discussion

### 3.1. Structure and Tribological Properties

The SEM morphology (FEI Company, Hillsboro, OR, USA) and composition of Mo/MoS$_2$-Pb-PbS films before and after UV radiation were showed by Figure 2. Before UV radiation, the Mo/MoS$_2$-Pb-PbS films was made up of small nano-scale particles that are superimposed on each other. The pores between the particles are large and the film structure is relatively loose. The large particles in the original Mo-Pb film are also broken into "cauliflower"-like agglomerates. There may be two reasons for the changes in the microstructure of the film. The reasons had been described in the reference [14] written by

our research team. After 40 h UV radiation, there were few micron-sized cauliflower-like large particles on the surface of the film. Most region of the surface were stacked with nano-sized small particles, and the macro roughness of the film was reduced. It was worth noting that some small particles with a size of hundreds of nanometers are arranged in a relatively regular pattern of "turtle shell" (as shown in the red dotted line area in Figure 2c). Combined with the element distribution on the surface of the film (Figure 3), the Pb content in the large particle area of the original film was higher, and the Mo content in the relatively flat area between large particles is higher. The morphology and composition of Mo/MoS$_2$-Pb-PbS composite films before and after UV radiation was shown by Figure 2. The scan area of EDS was marked with red rectangular box.

The analysis showed during UV radiation the large particles with high Pb and PbS contents on the original film surface were easy to decompose and break, while the gap area with high Mo and MoS$_2$ contents was relatively stable. After the large particles are broken, an amount of submicron small particles was scattered to the surrounding area and protrusions was formed, meanwhile the area where the original large particles are located was low-lying (shown in the red circle area). In summary, 'tortoise shell' pattern morphology was appeared gradually.

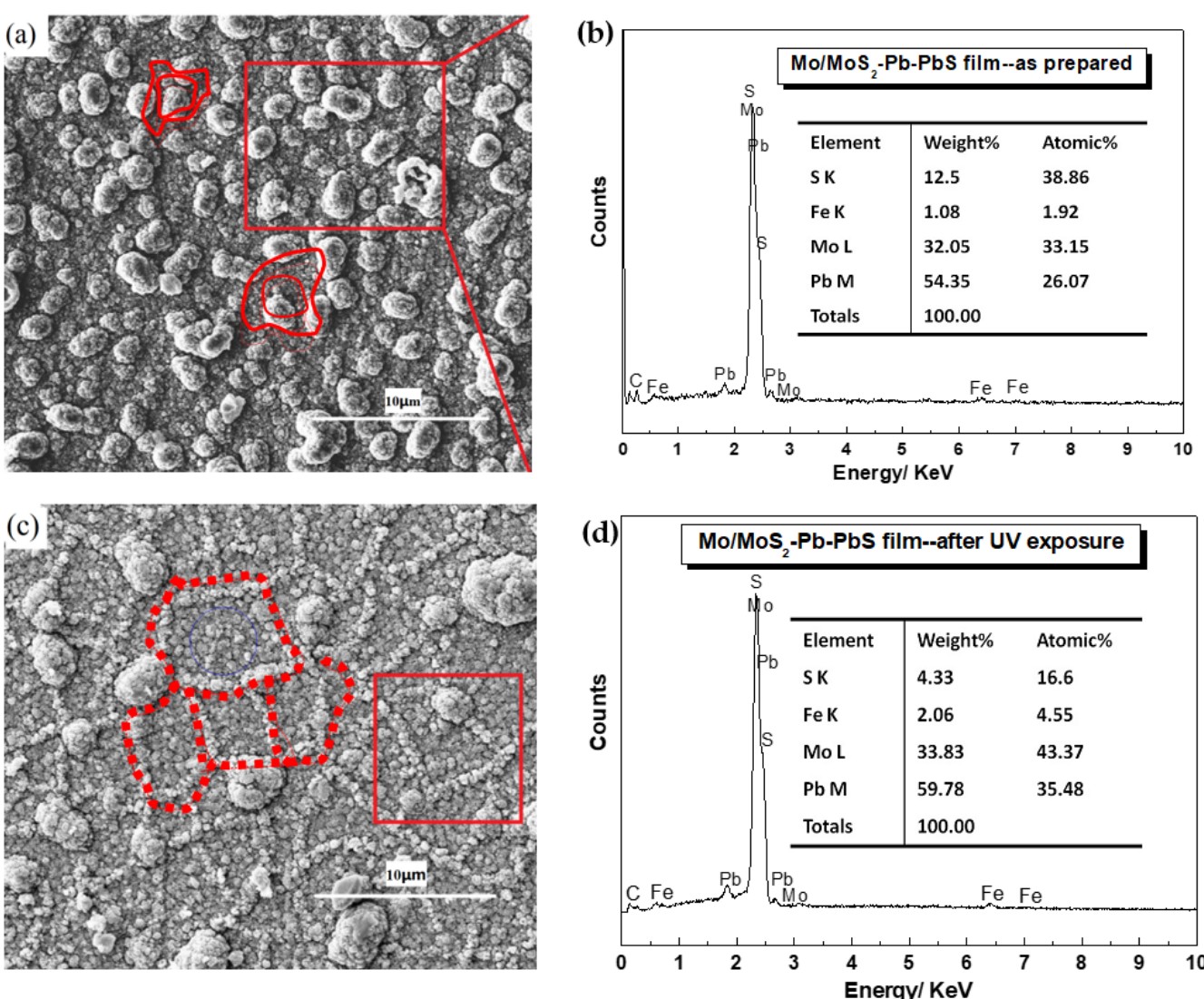

**Figure 2.** Morphology and composition of Mo/MoS$_2$-Pb-PbS composite films before and after UV radiation. (**a**,**b**) SEM and composition of Mo/MoS$_2$-Pb-PbS composite films without UV radiation, (**c**,**d**) SEM and composition of Mo/MoS$_2$-Pb-PbS composite films after 40 h UV radiation.

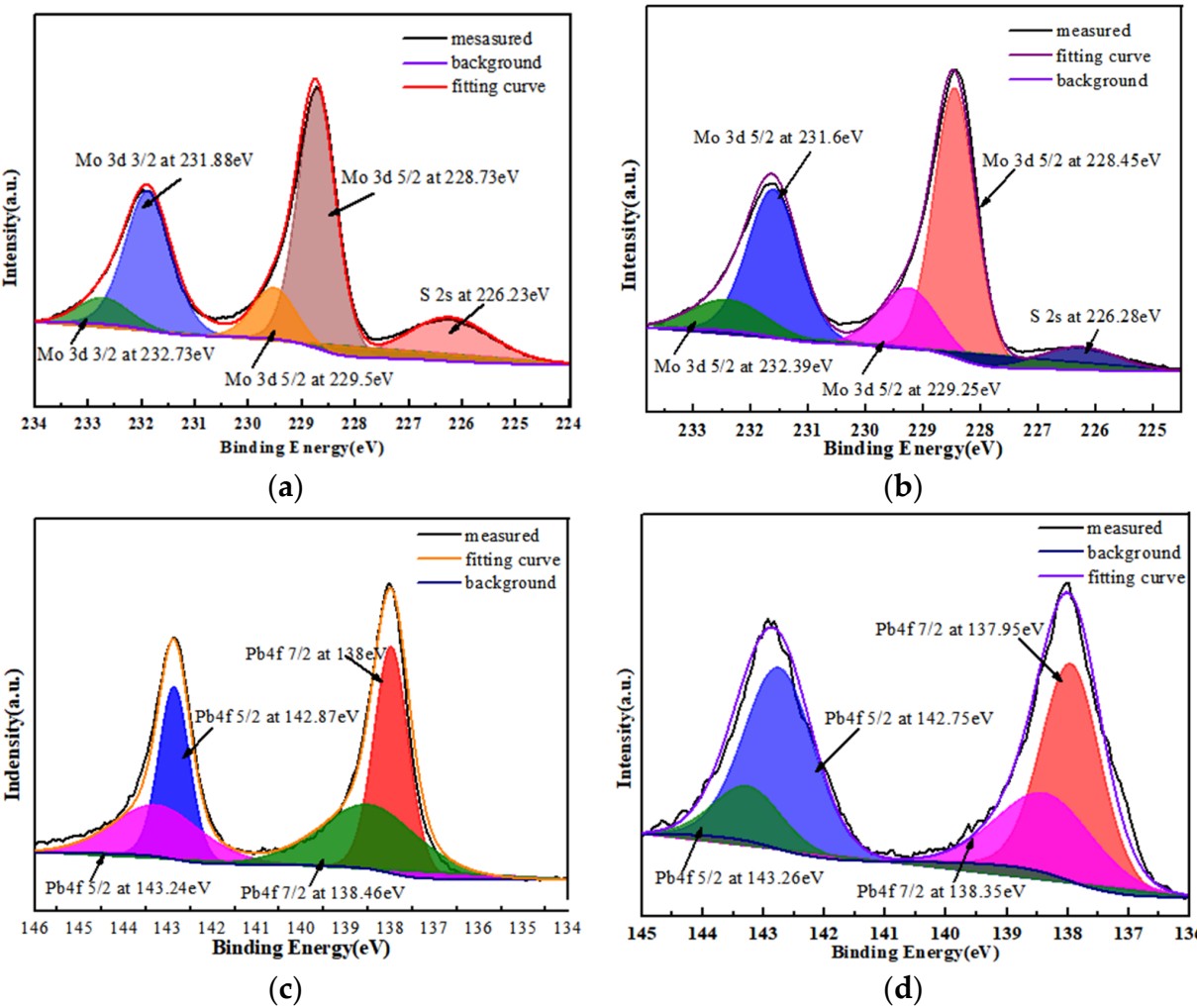

**Figure 3.** XPS of Mo/MoS₂-Pb-PbS composite films before and after UV radiation. (**a**) Mo3d without UV radiation; (**b**) Mo3d after 40 h UV radiation; (**c**) Pb4f-UV without UV radiation; (**d**) Pb4f-UV after 40 h UV radiation.

Before and after UV radiation, the elements on the film surface were the same. However, the content of S decreased from 38.86% to 16.6%, and the relative contents of Mo and Pb increased significantly, which indicated that UV radiation caused the loss of S on the film surface. The peak of Fe element spectrum was Fe detected in the substrate after X-ray penetration. The content of Fe element increases slightly after UV radiation, indicating that the thickness of the film decreased. The solid lubricating phase of metal sulfide (PbS and MoS₂) on the surface of the film has decomposed due to the increase in the content of elemental Mo and Pb after UV irradiation. The lubrication performance may be improved after UV irradiation.

Figure 3 showed the XPS analysis (VG Scientific, Windsor, UK) results of the valence state of Mo and Pb elements on the surface of Mo/MoS₂-Pb-PbS film before and after UV radiation. Mo element mainly existed in the form of MoS₂ and elemental Mo before and after UV radiation, but the relative content of MoS₂ decreased slightly and the content of elemental Mo increased after UV radiation. Pb element always existed in the form of PbS and elemental Pb, but the content of PbS on the film surface decreased and the content of elemental Pb increased after UV radiation. The valence changes of the two main elements indicated that the metal sulfide solid lubricating phases (PbS and MoS₂) on the film surface were decomposed during UV radiation. Combined with morphology and composition of Mo/MoS₂-Pb-PbS composite films before and after UV radiation, the surface roughness of

the decomposed layer decreased slightly, and the change of lubrication performance needs to be verified by friction test.

The comparison of the tribological properties of $Mo/MoS_2$-Pb-PbS films before and after UV radiation was showed in Figure 4. After UV radiation, the variation of the friction coefficient of the film was similar with curve of the original film, and the average friction coefficient of the film is finally stable at about 0.06. Nevertheless, the friction coefficient curves of the film were significantly different in the 'starting' and 'running-in' stages before and after UV radiation. After UV radiation, the starting friction coefficient of the film decreased, and the maximum value of the friction coefficient in the 'running-in' stage decreased from 0.17 to 0.12. However, the time of the 'running-in' stage was significantly prolonged, and the friction coefficient did not stabilize until 200 s.

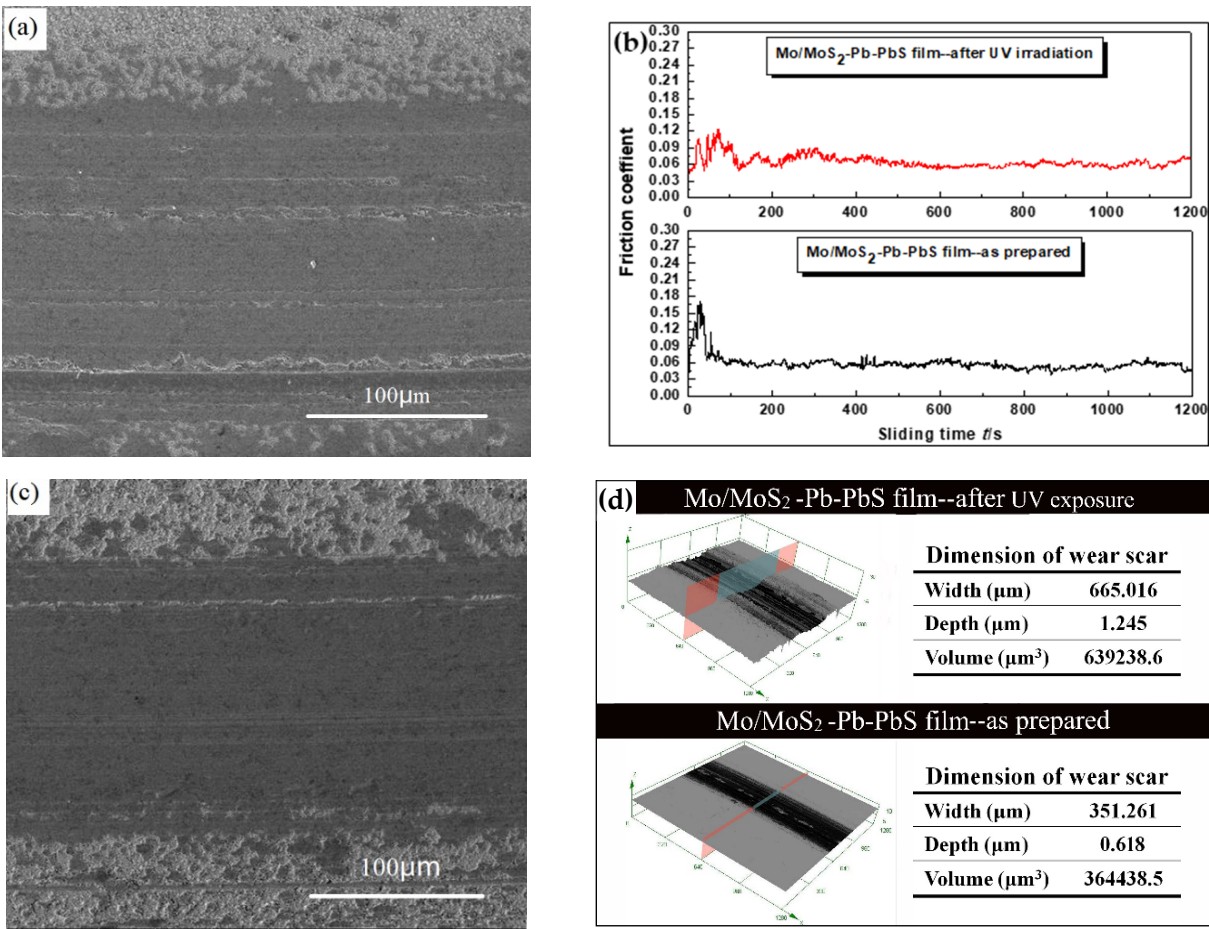

**Figure 4.** The tribological properties $Mo/MoS_2$-Pb-PbS composite film before and after UV radiation; (**a**) wear scar morphology without UV radiation; (**b**) friction coefficient curve; (**c**) wear scar morphology after 40 h UV radiation; (**d**) Three-dimensional wear scar morphology before and after UV radiation.

Before UV radiation, the wear failure of soft $Mo/MoS_2$-Pb-PbS films were dominated by plastic deformation and material transfer. When the friction test was carried out after UV radiation, the large-area plastic deformation and local material accumulation were significantly reduced. The friction track and the protruding particles around it were compacted and became smooth and dense. A small number of deep scratches was distributed along the sliding direction, and obvious material removal occurred in the central area of the wear scar. From the comparison of the three-dimensional morphology of the wear scars, the width, depth and wear volume of the wear scars on the film surface were significantly

increased, especially the large number of materials in the central area of the wear scar are removed to form a wide and deep groove after UV radiation.

Figure 5 showed the mass change and tribological performance comparison of Mo/MoS$_2$-Pb-PbS films after UV radiation at different times. UV radiation caused slight mass loss of the film. With the extension of radiation time, the mass loss of the film increased, and the mass change of the film tended to be stable after 60 h radiation. After UV radiation for different time, the average friction coefficient of the film did not change significantly in the stable period and remained around 0.06. When the radiation time was less than 40 h, the wear scar depth of the film increased linearly with the extension of the radiation time. When the radiation time continued to increase, the wear scar depth tended to be stable.

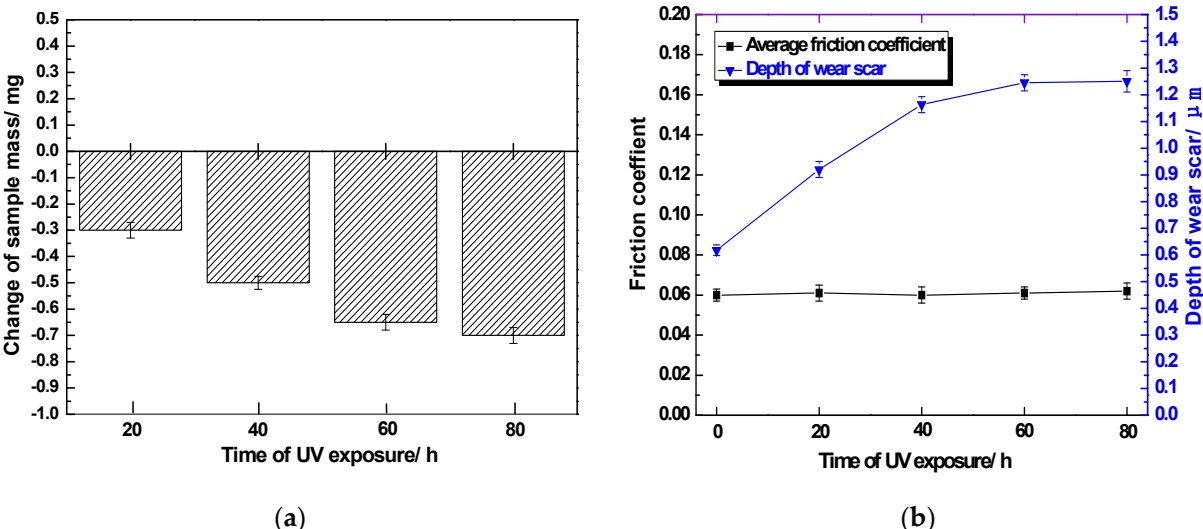

(**a**) (**b**)

**Figure 5.** The mass and tribological properties of Mo/MoS$_2$-Pb-PbS film after UV radiation for different time. (**a**) mass change; (**b**) average friction coefficient and variation of wear scar depth with radiation time.

In summary, the microstructure of the Mo/MoS$_2$-Pb-PbS films changed obviously after UV radiation. The micron-sized cauliflower-like large particles were broken in large quantities, and some small particles were arranged in the tortoise shell-like pattern. In addition, the key elements of lubrication components appeared obvious loss; the content of S element on the surface of Mo/MoS$_2$-Pb-PbS film decreased from 38.86% to 16.6%; so that there was a slight mass loss in the film. With the extension of UV radiation time, the film quality decreased, and the film quality loss increased with the increase of radiation time. The valence state of the main elements in Mo/MoS$_2$-Pb-PbS films did not change significantly, and the tribological properties of the films were degraded. After UV radiation, the fluctuation range of the friction coefficient of the film was significantly increased, the average value of the friction coefficient in the stable period was slightly increased so that the wear scar depth and wear volume were also increased accordingly. The average friction coefficient of Mo/MoS$_2$-Pb-PbS film did not change significantly with the UV radiation time, and the large fluctuation of the friction coefficient curve was limited to the 'start' and run-in stages. After a certain cycle of sliding, the friction coefficient tended to be stable again.

### 3.2. Damage Mechanism

From the summary and analysis of last section, UV radiation had indeed changed the microstructure and composition of Mo/MoS$_2$-Pb-PbS lubricating films, and the tribological properties had also been degraded to varying degrees. Studies had shown that the damage of UV radiation on spacecraft surface materials was mainly the photoelectric reaction or

photochemical reaction after the surface molecules of the material absorb the energy of high-energy ultraviolet photons. The ability of ultraviolet photons with a wavelength of 100–400 nm was enough to break the chemical bonds of most polymers and formed highly active free radicals on the surface [15,16].

After UV radiation, the content of S and MoS$_2$ in the composite films decreased significantly. The most important element Mo in the film existed in the form of Mo, MoO$_3$ and MoS$_2$ before and after UV radiation, and no new compounds are formed. MoO$_3$ is due to the oxidation of the film during the preparation of XPS samples. Pb in the Mo/MoS$_2$-Pb-PbS film did not undergo any chemical reaction. The analysis showed that the decrease of lubricating components and the tribological properties in the film may be due to some physical changes in the film during UV radiation.

Comprehensive analysis showed the ultraviolet beam with high energy density converges on the sample surface can produce high temperature because the radiation intensity of the ultraviolet radiation simulation device in this study reached 5 times the solar constant, and the deuterium lamp simulating vacuum ultraviolet (VUV) radiation is directly placed near the surface of the sample. In order to investigate whether the structural and performance changes of Mo-based multi-component composite films were related to the high temperature during the operation of the equipment, the Pt 100 platinum resistance was installed near the surface of the sample to test the temperature change during the UV radiation test especially, as shown in Figure 6.

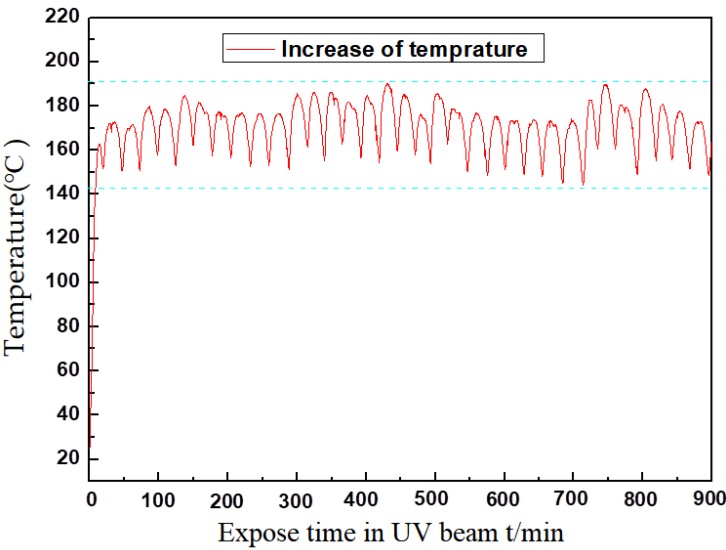

**Figure 6.** Temperature curve near the sample surface during the operation of UV radiation simulator.

At the beginning of UV radiation device working, the temperature near the sample rose sharply. After 13 min, the temperature reached 160 °C. When work time was 50 min temperature rose to 170 °C. During the left working time, the temperature near the sample fluctuated around 170 °C, and the maximum temperature reached 190 °C as shown in Figure 6.

The Mo/MoS$_2$-Pb-PbS composite films were prepared by two-step composite process of RF magnetron sputtering and low-temperature ion sulfurization from the review of the preparation process of solid lubrication films in this paper. During the low-temperature ion sulfurization process, the temperature was maintained at about 200 °C by the bombardment of ammonia ions, and most of crystal defects were induced on the surface of the film so that sulfur atoms (ions) diffused into the film and reacted to generate sulfides. The thickness of the sulfurized layer was generally micron level. With the extension of sulfurization time, S element will be enriched on the surface of the film and form a certain concentration gradient. The existence of S element enrichment layer on the surface was an important

reason for the thin metal sulfide film formed by sulfur infiltration to maintain good friction and wear resistance for a long time.

The non-bonding S element enriched in the surface layer of Mo-based multi-component composite solid lubricating film will be precipitated in large quantities, and a small amount of sulfide may also be decomposed under the action of local high temperature, so that the content of S element in the surface layer of the film decreased sharply, and the microstructure of the film changed. After 80 h UV radiation, the Mo/MoS$_2$-Pb-PbS film was reduced to 0.7 mg. The surface layer of Mo-based composite film not only lost S element, but also became very dry due to high temperature heating. Therefore, the starting friction coefficient of the film increased, and the fluctuation of the friction coefficient increased. However, the effect of UV radiation was limited to the most surface layer of the film. After a period of sliding, the tribological properties of the film would recover quickly.

### 4. Conclusions

After UV radiation, the microstructure of the films changed significantly. The relative content of S element and MoS$_2$ on the surface of the films decreased, and the mass loss of the films occurred to varying degrees. After long-term UV radiation, the friction coefficient fluctuation of the film at the starting stage increased. The tribological properties will also recover with the extension of sliding time; metals (Mo, Pb) and metal compounds (MoS$_2$, PbS) had shown good UV radiation resistance. The damage of Mo/MoS$_2$-Pb-PbS under UV radiation was mainly caused by the volatilization of the enriched S element in the surface layer due to the high temperature heating of UV radiation.

**Author Contributions:** Data curation, J.S.; Formal analysis, Q.Y.; Funding acquisition, C.H. and G.M.; Investigation, C.H. and Q.Y.; Project administration, G.M.; Resources, G.M.; Supervision, G.L.; Validation, H.W.; Writing—review & editing, Z.L. All authors have read and agreed to the published version of the manuscript.

**Funding:** The author acknowledges the financial and moral support and facilities provided by Hebei University of Technology, Army Academy of Armored Forces and Tianjin University of Technology and Education for the conduction of research. The research was funded by National Natural Science Foundation of China (Grant No. 52105200; 52122508; 51905533; 52075543).

**Institutional Review Board Statement:** Not applicable.

**Informed Consent Statement:** Not applicable.

**Data Availability Statement:** All data generated or analyzed during this study are included in this published article.

**Conflicts of Interest:** The authors declare no conflict of interest.

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
