# Peer review of "Effect of UV Radiation on Structural Damage and Tribological Properties of Mo/MoS2-Pb-PbS Composite Films"

_coatings, doi:10.3390/coatings12010100_

Round 1

Reviewer 1 Report

In this study, the authors tested the effects of UV radiation and on structural damage and tribological 2 properties of Mo/MoS2-Pb-PbS films. 

1) The article must be revised by a native speaker to correct all typos and some expressions. The term "electronic volts" is not correct - should be replaced with "electronvolts" or "eV". Page 1, line 44 "Yan etl" should be corrected to "Yan et al."
2) Test and provide EDX ELEMENTAL MAPPING - to confirm the uniform distribution of all elements across the selected area. Show a cross-sectional SEM image to confirm the uniform thickness formation. 
3) Please confirm the structural properties of the film by XRD before and after UV radiation. 
4) EDX is not so quantitative method to analyze the elemental composition, it is suggested to do XRF instead. 
5) Statistical information was not found. How many samples were tested per batch trial? 

Author Response

Dear reviewer:

Thank you for your reports about our paper, “Effect of UV radiation on structural damage and tribological properties of Mo/MoS2-Pb-PbS composite films” (coatings-1516070). These comments are all valuable and very helpful for revising and improving our paper, as well as the important guiding significance to our researches. We revised the manuscript in accordance with your comments, and carefully proof-read the manuscript to minimize typographical, grammatical, and bibliographical errors. We have made all changes into a red color with yellow bottom frame in the revised submission for easy tracking. A list of responses to  comments was attached as the supplemental material. The responds to the  comments are as follows.

If there were any questions, please contact with me.

Sincerely yours,

Han Cuihong

  • The article must be revised by a native speaker to correct all typos and some expressions. The term "electronic volts" is not correct - should be replaced with "electron volts" or "eV". Page 1, line 44 "Yan etl" should be corrected to "Yan et al."

Response: Thank you for your kind advices. We are sorry for our poor English writing and revise this paper as soon as possible. We have made all changes into red color and highlight with yellow in the revised submission for easy tracking. And we have had the manuscript polished with a professional assistance in writing.

  • Test and provide EDX ELEMENTAL MAPPING - to confirm the uniform distribution of all elements across the selected area. Show a cross-sectional SEM image to confirm the uniform thickness formation. 

Response: Thank you for your suggestions. We are sorry for our negligence of the positional relationship of EDX area. We revised the SEM picture and marked the scan area with red rectangular box. Since the UV radiation energy has little effect on the damage ability and damage depth of metal compounds, especially those with large binding bond energy, the distribution of film thickness has no significant effect on this study.

  • Please confirm the structural properties of the film by XRD before and after UV radiation. 

Response: Thank you for your kind advices. XRD can provide the structural composition of the material, the structure or morphology of atoms and molecules. XPS can anlysis element and determine valence of the film.

For Mo/MoS2-Pb-PbS composite films, there are three elements only, molybdenum, lead and sulfur elements. Molybdenum and lead elements are positive, and sulfur is negative. The formed compounds by them are relatively simple and obvious before and after UV radiation. Although only the valence states of elements were marked in the XPS results, the corresponding compounds can be easily determined. Therefore XRD is not very necessary in our research.

4) EDX is not so quantitative method to analyze the elemental composition, it is suggested to do XRF instead. 

Response: thank you for your kind advices. It is generally believed that EDS can determine the element content, although there will be 1 % measurement error. In this experiment, the content differences of Mo and Pb elements in Mo/MoS2-Pb-PbS composite film before and after UV irradiation were 1.8 and 4.4 %, respectively, which were greater than 1%. The increase trend of element content after irradiation is obvious, which is also our concern. If the element content measured by XRF is more accurate, we are very sorry that XRF cannot be completed on schedule due to Coronavirus and the arrival of winter vacation.

5) Statistical information was not found. How many samples were tested per batch trial? 

Response: Thank you for your good advices. We are sorry for our negligence of the samples. We had added this information in our paper.

There are eight samples exposed in UV environment, two samples as a group. The exposure time of each group was setting as 20 h, 40 h, 60 h and 80 h in order, which was equivalent to the radiation of the actual space exposure for 100 h, 200 h, 300 h and 400 h.

Reviewer 2 Report

The manuscript “Effect of UV radiation on structural damage and tribological properties of Mo/MoS2-Pb-PbS composite films” is a well-executed and nicely written piece of scientific work. The manuscript is suitable and may be accepted after minor changes. However, more literature needs to be discussed.

Author Response

Dear reviewer:

Thank you for your reports about our paper, “Effect of UV radiation on structural damage and tribological properties of Mo/MoS2-Pb-PbS composite films” (coatings-1516070). These comments are all valuable and very helpful for revising and improving our paper, as well as the important guiding significance to our researches. We revised the manuscript in accordance with your comments, and carefully proof-read the manuscript to minimize typographical, grammatical, and bibliographical errors. We have made all changes into a red color with yellow bottom frame in the revised submission for easy tracking. A list of responses to comments was attached as the supplemental material. The responds to the comments are as follows.

If there were any questions, please contact with me.

Sincerely yours,

Han Cuihong

Attachments of response

2#

  • The manuscript “Effect of UV radiation on structural damage and tribological properties of Mo/MoS2-Pb-PbS composite films” is a well-executed and nicely written piece of scientific work. The manuscript is suitable and may be accepted after minor changes. However, more literature needs to be discussed.

Response: Thank you for your good advices. We had tried our best to discuss the literature in the article. you can find the discussed part marked with yellow color.

Reviewer 3 Report

This work investigates ultraviolet (UV) radiation effects on tribological properties of Mo/MoS2-15 Pb-PbS film. In order to achieve this, the authors perform a series of ultraviolet (UV) radiation expose tests were carried out for 20 h,40 h,60 h and 16 80 h using a space UV radiation simulation device developed by their own team. The SEM morphology and composition of the Mo/MoS2-Pb-PbS films before and after 123 UV radiation  is explored. XPS and TEM techniques are also employed to characterize the samples.The paper is well structured, relatively well written and the results seem sound, so it is worth publication. The text must are reviewed as some misprints can be found. For instance, in fig. 2a and 2b, the scale bar is not visible.

Author Response

(The authors gave the same response as above.)

Round 2

Reviewer 1 Report

no more comments